# Assessment of Microplastics Pollution on Soil Health and Eco-toxicological Risk in Horticulture

Umesh Sharma [1], Sunny Sharma [2,*], Vishal Singh Rana [3], Neerja Rana [4], Vijay Kumar [3], Shilpa Sharma [4], Humaira Qadri [5], Vineet Kumar [6,7] and Sartaj Ahmad Bhat [8,*]

1  Department of Tree Improvement and Genetic Resources, College of Forestry, Dr. YSP University of Horticulture and Forestry, Nauni 173230, Himachal Pradesh, India
2  Department of Horticulture, School of Agriculture, Lovely Professional University, Phagwara 144411, Punjab, India
3  Department of Fruit Science, College of Horticulture, Dr. YSP University of Horticulture and Forestry, Nauni 173230, Himachal Pradesh, India
4  Department of Basic Sciences, College of Forestry, Dr. Yashwant Singh Parmar University of Horticulture and Forestry, Nauni 173230, Himachal Pradesh, India
5  Department of Environmental Sciences, J&K Higher Education Department, Govt. Degree College, Baramulla 193101, Jammu and Kashmir, India
6  Department of Basic and Applied Sciences, School of Engineering and Sciences, GD Goenka University, Sohna Road, Gurugram 122103, Haryana, India
7  Department of Microbiology, School of Life Sciences, Central University of Rajasthan, Ajmer 305817, Rajasthan, India
8  River Basin Research Center, Gifu University, 1-1 Yanagido, Gifu 501-1193, Japan
*  Correspondence: sunny.29533@lpu.co.in (S.S.); sartajbhat88@gmail.com (S.A.B.)

**Abstract:** In recent times, the existence of microplastics in the food chain has emerged as a physiological stressor and a multifactorial food safety issue, necessitating an immediate strategic perspective due to the associated human health and eco-toxicological risks. To the best of our knowledge, edible fruit crop facts have not yet been compiled, despite their presence in various food webs. Due to the risks associated with the public's health when consuming products (e.g., fruit crops) that contain a high concentration of microplastic pollutants, a strategic approach to the emerging issue is essential. In this review, we discussed the possible sources of microplastics and their effect on horticultural crops, soil, and microorganisms; the techniques used to know the constitution of microplastics; the eco-toxicity of microplastics and their derivatives on horticultural crops; and suggested mitigation measures and public policies on control alternatives. This research aims to help environmentalists, biotechnologists, and policymakers understand the mechanism and dynamics of microplastics in soils and edible parts so that potential risks can be mitigated in advance.

**Keywords:** food systems; horticulture; soil health; microparticles; mitigation strategies

## 1. Introduction

Agricultural systems, particularly the horticulture industry, are contaminated with a wide number of pollutants, of which microplastics (MPs) also make major contributions, but their potential impact on soil health, eco-toxicology, and nutrient dynamics is relatively unknown [1]. In general, developing countries spread or dispose of a huge amount of plastic waste [2]. The most recent contaminant that surrounds rivers and enters the sea by runoffs is plastic (micro and nanoplastics), which makes up about 80% percent of the debris in the sea [3–5]. Overall, 6300 million tons (Mt) of plastic products are produced globally, of which more than 75% are deposited in landfills and various ecological segments. In 2017, plastic production worldwide was 368 Mt where China contributes 114 Mt followed by Europe (59 Mt), and its production capacity is expected two folds in 2040 [2]. The overuse of plastic commodities in industry and by global citizens has had an irrevocable influence

on terrestrial ecosystems. Researchers have investigated how microplastic particles go into the food chain and are consumed by humans, potentially endangering their health [3–7]. Therefore, an urgent study is required in the area of food security based on the sources of pollutants and substances used in horticulture businesses, as well as the establishment of rigorous public policies on alternative control methods.

The existence of microplastics from various sources in the terrestrial ecosystem has recently become a hotspot of current research [3]. The majority of microplastics (MPs; < 5 mm) found in nature are non-biodegradable and remain for a very long period. Due to the rapid increase in industrialization, plastic production has amplified exponentially; however, the management of waste material has a deficit thing due to the proliferation of microplastic pollution [6]. According to the World Health Organization (WHO), eating a fruit and vegetable-rich diet is the best strategy for maintaining good health [7]. However, fruits grown in contaminated soil are unclean and pose health dangers to people. As a result, it's critical to regularly check the quality of the food and consider the significant prevalence of microplastics in the atmosphere. Researchers found that the assessment of microplastics and nanoplastics (NPs) is a highly relevant and widespread topic of discussion due to the lack of strong national and international rules or food standard limits for plastic contamination control [7–10].

Recently, microplastic pollution in horticultural science contributes a huge share to the environment and generated irreversible global impacts on the natural environment. Generally, microplastic wastes reach terrestrial soil through runoffs and uptake by the roots and are translocated to various plant parts such as leaves, stems, and fruits [9,10]. Despite this, numerous plastic implements used in the horticulture industry would likewise introduce microplastics to either horticultural products or human consumption products [11]. However, its impacts have not been proven in a consistent manner, and experimental circumstances in the published literature vary. In addition, the uptake and translocation mechanisms of several edible species have received minimal research [12]. In addition, they stated that the procedures utilized to determine the size of the particles, source, and amount of organic material (OM) in edible organs differ from microplastic variations.

Literature reveals a lack of proper understanding of the destiny and consequences of nanomaterials in horticulture systems, which is problematic considering the microplastics' potential risk to human health and their uptake through the food web. In the context of microplastic pollution, it is a rapidly emerging subject for nanotechnology research, e.g., nanoplastics. Microplastics of all forms and sizes can infiltrate the root, seed, leaf, culm, and plant cells of fruit crops [10–12]. The size of microplastics alone is of foremost importance to understanding the food web and health risk assessment. The contamination through microplastics in the horticultural sector worldwide is a fact. In this review, we discussed the possible sources of microplastics and their effect on horticultural crops, soil, and microorganisms; techniques used to know the composition of microplastics, the eco-toxicity of microplastics and their derivatives on horticultural crops, and suggested mitigation measures and public policies on control alternatives.

## 2. Classification of Microplastics

Professor Richard Thompson coined the term "microplastics" in 2004 [13]. Microplastic quantification and classification are useful criteria to monitor the magnitude of pollution created by microplastics and the potential health hazards to living organisms. Globally, the use of plastics is increasing rapidly, with the amount of usage having more than doubled in the last two decades. Plastic waste comes from packaging, which accounts for 40 percent, consumer products for 12 percent, and apparel and textiles for 11 percent [14]. As per the latest data, 55 percent of plastic garbage worldwide was thrown in local vicinities and soil while 25 percent was burned, and 20 percent was recycled [2]. Primary plastics, secondary plastics, and nano-plastics are the several groups into which microplastics are separated (Figures 1–3).

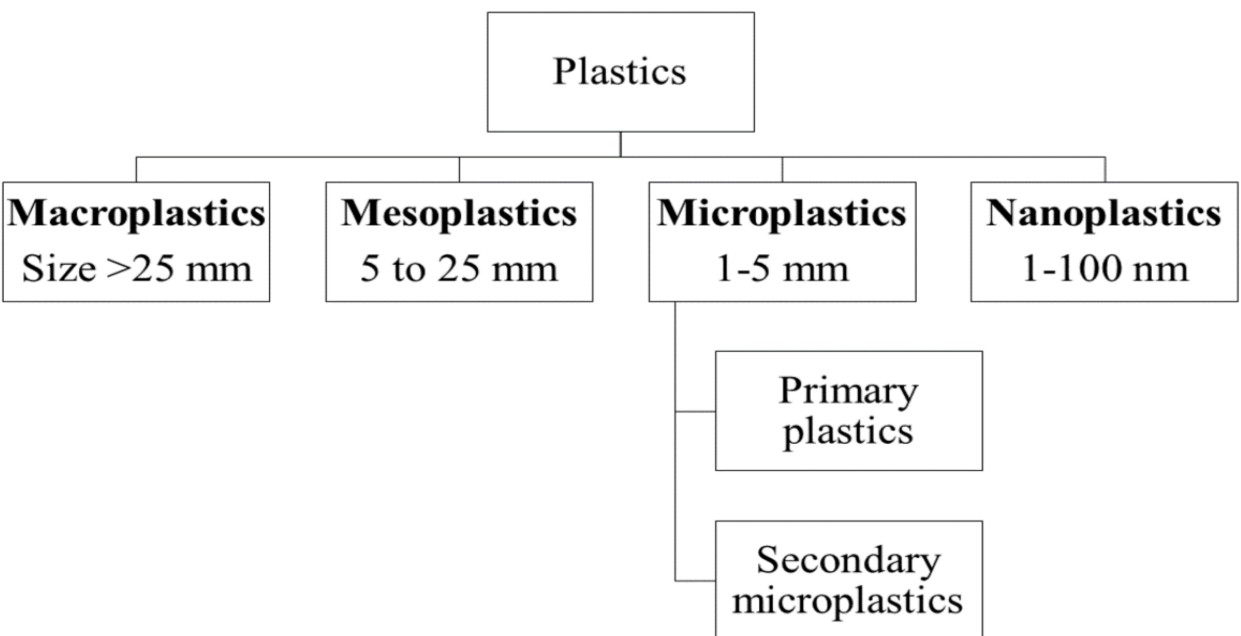

**Figure 1.** Various classifications of the plastics based on size.

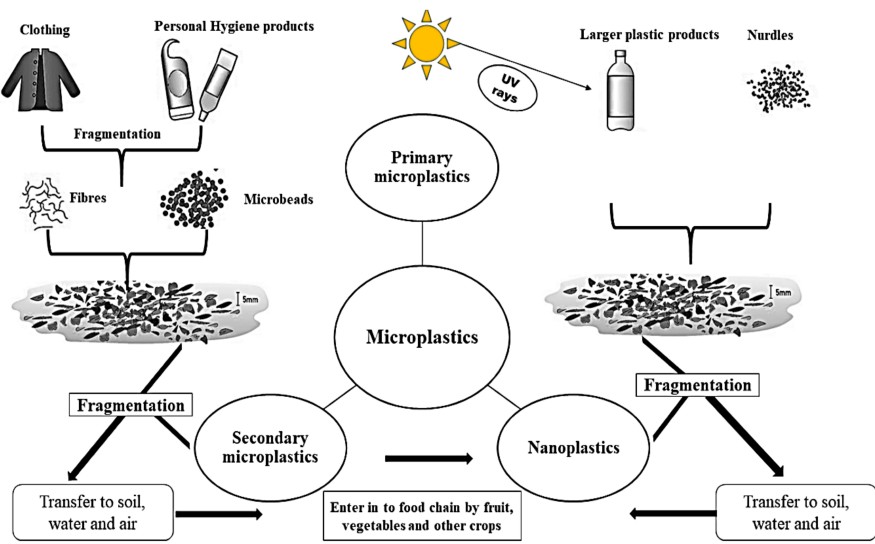

**Figure 2.** Microplastics classification, their sources, and transformation, and transport.

### 2.1. Primary Microplastics

These types of plastics are tiny in size and are used for various industrial processes, namely, housing material and transportation spare parts, cosmetics, clothing, fishing nets, and medicinal purposes. It is normally synthesized using polyethylene and polystyrene. Pollution through microplastics mainly comes from the packaging of personal care products, pellets, electronic devices, vehicles, or printers [2,13–16]. For example, earlier, we used natural scrubbers that are made from almond shells, oatmeal, and pumice, but due to modernization, microplastic "scrubbers" are used to a greater extent. These microplastics are creating pollution because bioplastic microbeads have a long degradation period as compared with normal plastic. This microplastic is also used in the air blasting technique [7].

### 2.2. Secondary Micro-Plastics

Secondary microplastics are a form of plastic that is generally derived from the fragmentation of larger pieces and gets converted into small pieces both at sea and on land.

Fragmentation of larger plastics can be accomplished by photo-degradation caused by electromagnetic radiation. This can reduce the shape and structural integrity of microplastic particles. This microplastic is unnoticeable to the naked eye [17,18]. In the ocean, secondary microplastics are detected at a size of 1.6 μm in diameter. Microplastic fibers enter the environment from the synthetic clothes [19]. Similarly, tires are made from styrene-butadiene rubber that is reduced into tiny plastic particles. Furthermore, microplastics plastic pellets used to create other plastic products [20]. Microplastic pellets are also utilized to make additional plastic goods [20,21].

*2.3. Nanoplastics*

Nanoplastics are the third type of plastic pollution that is less than 1 μm in size [17,20]. Various reports showed that nanoplastics in the atmosphere might be an environmental threat due to their continuously rising concentrations [22]. The determination of nanoplastics' quantity in the environment might be possible with various tools, such as spectroscopy [23]. Spectroscopy is a novel technique for the identification and quantification of nanoplastics [24,25]. These types of plastics may pose a risk to the terrestrial atmosphere because the small size of microparticles allows them to easily cross cellular membranes and disturb the functioning of living organisms [26]. The nature of nanoplastics is lipophilic, which may bind plastic particles with the core of lipid bilayers [25–30]. The study showed that the nanoplastics have surpassed the epithelial membrane of fish, and microparticles get deposited into various organs, namely, the gall bladder, pancreas, and the brain [29]. Limited literature is available on the bad effects of nanoplastics on the human body. In other organisms, such as fish or other aquatic organisms, nanoplastics can be increasing the stress response pathway that changes sugars and cortisol levels [29].

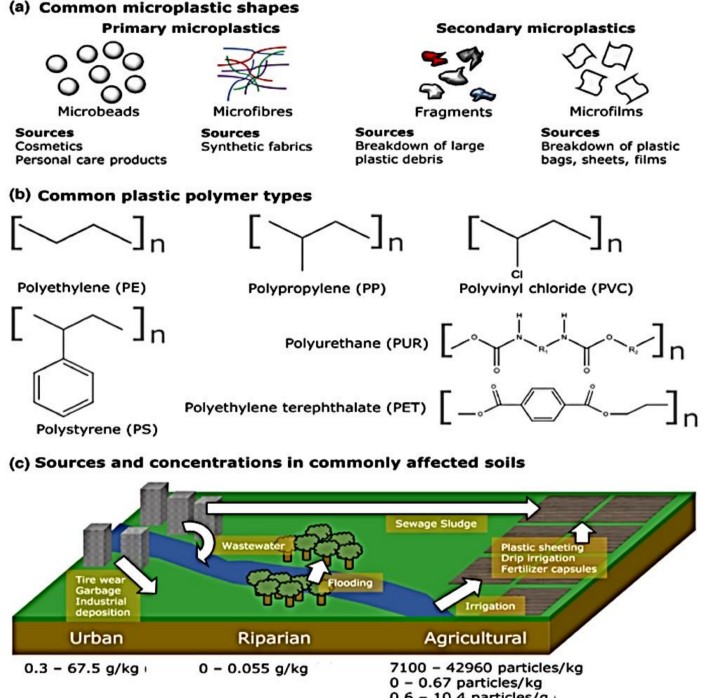

**Figure 3.** Microplastic forms, shapes, and origins, as well as microplastic concentrations in urban and agricultural soils Reprinted from Helmberger et al. [29] with permission of © (2019) Wiley.

## 3. Microplastics Consumption and Their Eco-Toxicological Risk

Food safety is a major concern with an increasing population. Healthy feed is primarily a global challenge because food adulteration is started from sowing to consumption. Various reports have shown that the intensity of microplastics pollution is gradually in-

creasing worldwide [17,18]. Thus, the ill effect of microplastic pollution on human health, as well as the orchards, has been understudied. It can also be noted that microplastic pollution has disturbed our ecosystem by adulterating products either from marine life or standard products of daily needs, namely, fruits, vegetables, table salt, sugar, honey, and even packed water [31]. Another report showed that vegetable fields were contaminated with microplastics in China. It was found that there is a presence of pollutants in vegetables [32].

Recently, various reports showed the different types of plastics and their derivatives, namely, PVC, polystyrene, and phthalates are possibly carcinogenic [17,18,20,25,32]. However, the physico-chemical impacts of these derivatives in the human body relate to the mobilization of some other pollution derivatives, such as polychlorinated biphenyls (PCBs) and dichlorodiphenyltrichloroethane (DDT) [33]. In other words, people are constantly unsafe from primary and secondary microplastics due to the continued use of these products. Products such as cosmetics can enhance the spontaneous absorption of pollutants derivates which can damage skin hence, leads to cause of skin related problems. In addition to that, these derivatives are having ill impacts on human health that can mainly depend on the concentration of use and 90 percent of these plastics derivatives are gulped through the fecal route [31–33].

The consumption of different microplastics depends on various factors such as age, sex, concentration, and environment [34]. However, the impacts on health are dependent on the size, polymer type, and additive derivatives of primary or secondary microplastics ingested by people [35]. While there is only limited literature available on the accumulation of microplastics and their long-term effects on the human body. It has been reported that the accumulation and oral exposure to microplastics has some negative effects on the human body's inflammatory responses and alters the intestinal microbiome cardiopulmonary, metabolites balance, and reactive oxygen species (ROS) [34]. The long-term impacts of microplastic particles on aquatic and terrestrial ecosystems are still unknown. Hence, the research on pollutants and their ill effects on ecosystems should be a priority for developing mitigation strategies in the context of food security. The recent literature showed that microplastic effects on different edible crops have been summarized in Table 1 and Figure 4. The *Lemna minor* L. wild species are used in the food industry and were exposed to polyethylene and two different kinds of exfoliating microbeads. It was found that the presence of microparticles causes mechanical stress, resulting in a reduction in root growth proliferation [36,37]. Conti et al. [17] also exposed and evaluated different numbers and sizes of microplastics in various fruits and vegetables. After the extraction by using an innovative SEM-EDX, it was estimated that higher levels of microplastics were found in apples and carrots. Considerably, lower levels of microplastics were observed in lettuce samples. The smallest and biggest microplastics in size were noticed in the carrot and lettuce samples.

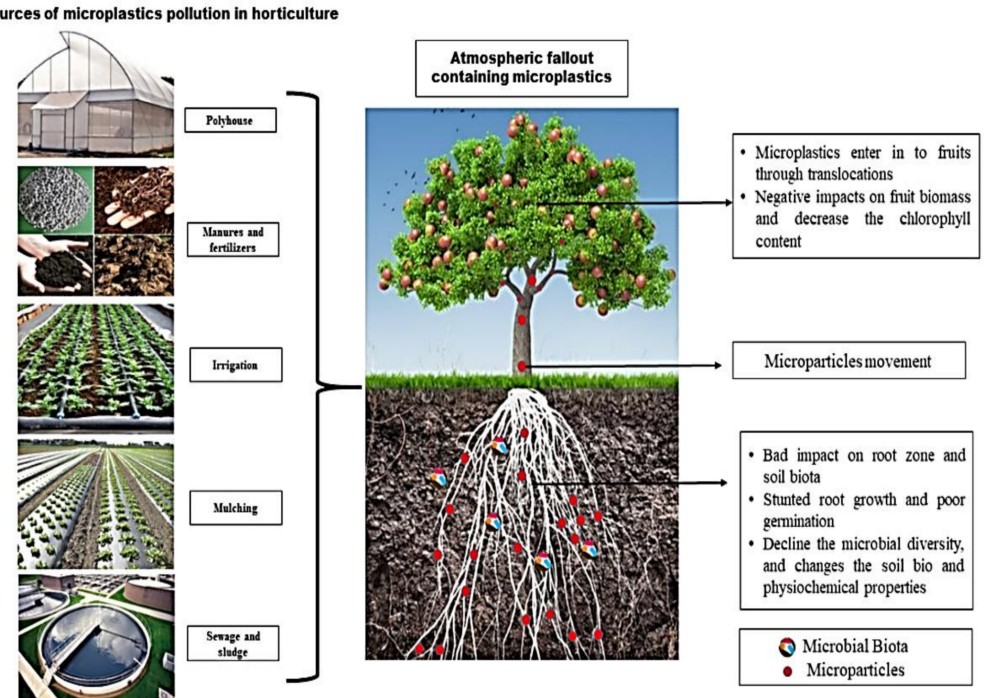

**Figure 4.** Possible sources and their effect on crops, soil, and microorganisms.

The mechanism of absorption and translocation of microplastics is nearly identical to that of carbon nanomaterials. The potential path of microplastics from the environment to vegetables allows for translocation or uptake inside their biological systems. Li et al. [38] reported on the toxicological effects of microplastics on vegetables. They used two types of polyvinyl chloride microplastics at varied concentrations of 0.5, 1, and 2 percent. The results showed that there was a significant difference in the activity of roots in terms of root length, diameter, and volume surface area. They also reported the effect of microplastics on the leaves and revealed a reduction in the absorption capacity, capture, and transfer of light energy electrons.

Li et al. [16] also studied the impact of several microplastics, including polyester fibers, polyamide beads, polyethylene, and polystyrene fragments, on chives. Similarly, Ng et al. [15] showed the effect of nano- and microplastics and revealed that an accumulation of 50 nm-sized microplastics in the seed coat of garden cress retards water uptake activity, or imbibition, which reduces the germination and root growth when exposed to plastic particles with a size of 50 nm. Li et al. [16] also found that polystyrene nanoplastics accumulated in the cucumber roots. Moreover, these particles were shifted from the root to the above-ground parts such as stems, leaves, flowers, and fruits and ameliorated the amount of soluble protein and reduced the concentration of calcium, magnesium, and iron.

Due to the availability of microplastics' pollutant derivatives in irrigation sources, such as sewage sludge applied to agricultural fields or riverine or pond water, microplastics also pose a rising hazard to environmental conditions and horticultural productivity (Table 1). Hernández et al. [39] examined the effect of sewage sludge containing microplastics on the growth and yield of tomato plants by exposing tomato plants to sludge containing microplastics in Murcia, Spain. They investigated and calculated that sewage sludge contains up to 31,100 particles per kilogram of dry weight, which retards the growth and productivity of tomato plants and delays fruit production. Similar to irrigation water, little is known about the eco-toxicity of microplastics in water [19]. In addition, Calderón-Preciado et al. [21] estimate that people are exposed to various forms of microplastics through the intake of fruits and vegetables. The irrigation sources, such as rivers or sewage water, are also the main sources of soil and crop contamination. This highly diverse nature of the rivers, which can have microplastic derivatives, cannot be easily checked.

**Table 1.** Eco-toxicity of microplastics and their derivatives on different horticultural crops.

| Crop | Country | Doses | Type of Plastic | Effects | References |
|---|---|---|---|---|---|
| Duckweed *Lemna minor* | Slovenia | — | Polyethylene microbeads | <ul><li>Microbeads maintained the growth and chlorophyll content</li><li>Retard the root growth due to</li><li>mechanical blocking</li></ul> | [36] |
| Garden Cress *Lepidium sativum* | Netherlands | 50 nm, 500 nm and 4800 nm plastic particles | Green fluorescent plastic particles | <ul><li>Reduce the germination percentage after exposure for 8 h</li><li>After 24-h exposure, the germination rate</li><li>did not differ significantly.</li></ul> | [40] |
| Broad beans *Vicia faba* | China | Root tips 10, 50, and 100 mg/L of microplastics of 5 μm and 100 nm. | Polystyrene fluorescent microplastics | <ul><li>Decrease the root biomass and</li><li>catalase activity</li><li>Increased superoxide dismutase and peroxidase activities.</li><li>Oxidative damage</li></ul> | [41] |
| Plant generally Type | Germany | | Beads and fragments, Fibers, Films, Biodegradable Nanoplastics | <ul><li>Changes in the plant growth</li><li>Alteration of soil structure and bulk density</li></ul> | [42] |
| Lettuce | China | 0.2 and 1.0 μm | Polystyrene microbeads | <ul><li>Disturb the nutrient uptake Tracking of uptake</li><li>Roots trapped, absorbed, and transported microplastics to stems</li><li>and leaves</li></ul> | [16] |
| *Lolium perenne* (perennial ryegrass) | United Kingdom | 1 g per kg of dry soil of polylactic acid (65.6 μm) and HDPE (102.6 μm) | Biodegradable polylactic acid and HDPE | <ul><li>Decrease the germination</li><li>Decrease the Root biomass of perennial</li><li>No significant variation in chlorophyll content</li></ul> | [12] |
| Onion *Allium cepa* | India | 25, 50, 100, 200, 400 mg per L and 100 nm | Micro-polystyrene | <ul><li>ROS generation and chromosomal abnormalities</li><li>lower the cdc2 gene expression</li></ul> | [43] |
| Duckweed *Lemna minor* | Ireland | — | Polyethylene microspheres | <ul><li>Absorb the more microplastics</li><li>Photosynthetic efficiency and</li><li>growth maintained</li></ul> | [44] |
| Tomato *Solanum lycopersicum* | Spain | Pot experiment | Sewage sludge containing microplastics | <ul><li>31,000 particles per kg dry weight were detected</li><li>Reduced the growth of tomato plants</li><li>Delayed and diminished fruit production.</li></ul> | [39] |
| Arabidopsis *Arabidopsis thaliana* | China | 0.3,1.0 g per kg and 10, 50,100 μg per Ml for soiland MS media, respectively | Polystyrene nanoparticles | <ul><li>Accumulation of microplastics in the root zone as well as observed in the apoplast and xylem</li><li>Increase the production of ROS and inhibitory effects on the growth of the plant</li></ul> | [45] |
| Carrot *Daucus carota* L. | China | PS: 10 and 20 mg per L 1, 2, and 4 mg per L 0.2 μm, 1μm | Polystyrene in hydroponic solution | <ul><li>Accumulation of microplastics into roots and translocated into leaves and root</li><li>Negative effects of polystyrene were found</li></ul> | [46] |
| Cucumber *Cucumis sativus* L. | China | 50 mg per L 100, 300, 500, and 700 nm in solution | Polystyrene nanoparticles | <ul><li>Accumulation of particles in the leaf, stem, flower, and fruit of the cucumber</li><li>Increase Malondialdehyde (MDA) and proline content in the root</li></ul> | [47] |
| Peas *Pisum sativum* | Korea | MP-contaminated soils 40, 20 mg/kg | Polystyrene | <ul><li>Reduce the plant yield</li><li>Significant changes in weight, colour, amino acids, and protein</li><li>Nutrient content increased</li><li>Microplastics infiltrated into incomplete casparian strips during root development</li></ul> | [48] |

Horticultural crops are herbaceous and perennial in nature and might be susceptible to microplastic contamination. Fruits and vegetable crops are a part of a healthy diet [18]. Therefore, microplastics are a significant problem, and urgent need exists to conduct toxicological and epidemiological research to study the potential impacts of microplastics on both human and plant life. The farm's inputs can also increase microplastic pollution, so checking the system of cultivation is also a point of concern.

## 4. Determination of Microplastics

Microplastics are present in the diverse nature of the environment and their potential nature can be determined with the quantification, separation, and classification methods. Determination of microplastics assists to check the vulnerability of microplastic pollution and potential damage risks to living biota. Microplastic in the terrestrial ecosystem is degraded by different mechanisms, namely, photo-oxidation, hydrolysis, biological degradation, and bio-assimilation by microorganisms [49]. The UV ray's photo-degradation can release reactive species to a greater extent. Microplastics from various sources with diverse chemical compositions and shapes are imaged by scanning electron microscopy. The plastic is varied in particle size, i.e., nanoparticles (1 to 100 nm), microplastics (1 to 5 mm), mesoplastics (5 to 25 mm), and macroplastics (>25 mm) in diameter [32,34,39,44,49]. Out of these plastics, micro- and nanoplastics are the most effective due to their high potential for bioaccumulation. The size and surface properties of microplastics can also help provide some information about their origins and products. The estimation can be acquired with various tools and methodologies. Table 2 shows the different technological interventions used for estimating the microplastics in horticultural commodities such as fruits, vegetables, and irrigation sources.

Most techniques are generally non-destructive and follow the physical determination of microplastics. Different approaches have been discussed and summarized in Table 2. The non-destructive approach can be accomplished by separating the microplastic derivatives from the samples, and the most common reagents are hydrochloric acid (HCL), nitric acid ($HNO_3$), potassium hydroxide (KOH), sodium hydroxide, alcohols, and hydrogen peroxide [39,50]. The various reagents listed here serve various purposes for the polymer in contact with the matrix. The sodium hydrochloride (NaCl) in the solution helps in separating the microplastic derivatives with a density between 1.2 to 1.9 g per $cm^3$ [13]. Similarly, potassium iodide, ethyl alcohol, and zinc chloride are alternative reagents that can be used for the determination process. Hydrogen peroxide ($H_2O_2$) is used to liquefy organic derivates and acts as an oxidative source that helps to cleanse the sample [51]. Digestion through enzymes is costlier than other techniques and destructive types of approaches [52,53].

**Table 2.** Estimation of microplastic and their constituent by using different techniques in horticultural crops.

| Type | Technology | Size/Quantity | Commodity | References |
|------|-----------|---------------|-----------|-----------|
| Polystyrene nanoplastics | SEM and LCSM | 100, 300, 500, and 700 nm | Farms soil | [54] |
| Polyvinyl chloride | Electron microscopy | PVC-a100 nm to 18 um PVC-b-18 to 150 um | Lettuce (*Lactuca sativa* L.) | [38] |
| Microbeads | IRMS | —— | Maize grown in hydroponics | [55] |
| Microplastics | SEM | 195,500 microparticles per gram | Apple *M. domestica* | [17] |

**Table 2.** *Cont.*

| Type | Technology | Size/Quantity | Commodity | References |
|---|---|---|---|---|
| Microplastics | SEM and EDAX | 189,550 microparticles/g | Pear *P. communis* | [17] |
| Microplastics | SEM and EDAX | 126,150 microparticles/g | Brocooli *B. oleracea italica* | [17] |
| Microplastics | SEM and EDAX | 50,550 microparticles/g | Lettuce *L. sativa* | [17] |
| Microplastics | SEM and EDAX | 101,950 microparticles/g | Carrot *D. carota* | [17] |
| Fibers and microbeads. | Microscope | <0.2 mm Polyamide (32.5%) and polypropylene (28.8%) | Vegetables | [32] |
| PVC, Nylon, Polystyrene | FTIR | 1 mm to 1.5 um | Sewage/sludge | [56] |
| Polystyrene nanoplastics | SEM | 100, 300, 500, and 700 nm | Cucumber plants | [47] |
| Small polystyrene | SEM | 100–1000 nm | Lettuce (*Lactuca sativa* L.) | [50] |
| Large polystyrene | SEM | Greater than 10,000 nm | Lettuce (*Lactuca sativa* L.) | [50] |
| Microfibers, HDPE, and LDPE | Raman spectroscopy | Microfibers, HDPE, and LDPE | Tomato field 0.4–2.6 mm | [38] |

SEM-Scanning Electron Microscope; FTIR-Fourier-Transform Infrared Spectroscopy; EDAX: Energy Dispersive X-Ray; IRMS- Isotope Ratio Mass Spectrometry.

## 5. Impacts of Microplastics on Soil Health

Soil is also an important part of the environment which can be contaminated by microplastics [27,32,33]. Soil health is often linked with bio and physico-chemical properties which can directly or indirectly regulate plant health. The physical, biological, and chemical properties are the key factors of soil health [57]. Fluctuations in the composition of any properties might affect crop growth and yield. Piehl et al. [58] quantified the 34±0.36 microparticles per kg of topsoil in a farmer's field even though he did not apply plastic to the field. Plastics use can also be extended up to the animal production systems [59]. The various direct sources of microplastics can be used in the horticultural system, i.e., greenhouses or polyhouses, polytunnels, covering sheets, mulching sheets, protective or anti-hail nets, coated fertilizers and pesticides, irrigation and dripline pipes, instrument nozzles, and drip systems [60]. The indirect sources of microplastics in agricultural soil such as sewage sludge and bio solids compost, manure, irrigation, and unauthorized dumping and oversight (Table 3). These sources generally affect the properties of the soil and especially disturb the microorganisms.

Microbial diversity is a boon for soil health, which is directly linked to the output of crops. There is a significant direct relationship between microbial diversity and yield in the arid zone [8,13,18,30,39,44,49,60]. Moreover, horticultural production can be maintained with the different strains of microorganisms that ameliorate soil health and improve long-term production [30]. However, microplastics can alter the biological as well as physical characteristics of the soil. Microplastic pollution also has an impact on several physicochemical and biological characteristics, such as cation exchange capacity, electrical conductivity, bulk density, water holding capacity, and microorganism interactions with $H_2O$ stable soil structure. The types and concentrations of microplastics have an impact on how they affect soil [61]. Alterations in soil microbial communities caused by the pres-

ence of microplastics also affect plant health, and the effect is likely to be negative if root symbionts such as mycorrhiza and nitrogen fixers are disrupted. The slow breakdown of microplastics has been connected to microbial immobilization, although the empirical proof for immobilization is currently lacking [41]. Furthermore, microplastics may serve as media that introduce phytotoxic substances into the soil, thus adversely affecting plant roots and health [53]. Generally, by altering soil structure and microbial diversity, microplastics could alter plant diversity and community composition. Nonetheless, while postulates were made by relating alteration in soil biophysical properties to the impacts of microplastics on plants, there are few studies to prove the postulates. In addition, microplastics may be used for the emergence of phytotoxic substances into the soil, negatively impacting plant health [8,12,53]. In general, microplastics could alter plant diversity and community composition by altering soil structure and microbial diversity.

In comparison to polyethylene fibers, polyester fibers have the potential to increase the maximum water-holding capacity (MWHC) of soil. Polyester fibers, on the other hand, increase the concentrations of polyacrylic and polyethylene microparticles, which decrease the bulk density of the soil [44,49,61] The extents of changes in soil properties are generally not proportionate, with stronger impacts induced at low microplastic concentrations and the increase in effects not correlating consistently with microplastic concentrations. These changes show that microplastics can have the potential to change the soil's biological and physical properties. Microplastics can also have the capacity to alter the activities of microorganisms, and this alteration is based more on concentration than the type of microplastic [62]. de Souza et al. [61] revealed that microplastic shape might play an important role in the activities of microorganisms. The linear microplastics, namely, polyacrylic and polyester, reduce microbial activities in comparison to the nonlinear plastics.

The activity of soil microorganisms changes with soil aggregation, similar to peds or clods, which can be affected by the microplastics [61,62]. This might be due to changes in the metabolic rates of microorganisms because of microplastic availability in the soil [63]. The nutrient profiling was also affected by the different derivatives of the microplastics. An addition of microplastics (28% $w/v$) to soil can ameliorate nutrient profiling [62]. This might be due to the enhancement in the activity of phenol oxidase enzyme and diacetate hydrolase, which can reduce the high-molecular weight to a lower one and improve the available nitrogen and phosphorus [64]. On the contrary, microplastics can also reduce the availability of C, N, and other nutrients [65].

Various studies have shown sewage and wastewater treatment sludge can also have a capturing power towards the microplastic captures. It was estimated that nearly 50 percent of sewage sludge was used as fertilizer sources in developed regions such as Europe and North America, resulting in a high amount of microplastic derivatives in the productive land [66]. Mulching, poly houses, and poly tunnels act as sources of microplastic introduction in the farmer's field [67]. In the field, plastic mulch is used to improve water use efficiency and help maintain the soil temperature [68]. However, a small piece of plastic that remains after the growing season can be a threat to the soil health.

Farmers frequently use plastic mulch, which increases the amount of plastic waste in the soil and raises the risk of it spreading to other ecosystems. Scientists have recently issued cautionary statements against the usage of plastic mulches in agriculture, claiming that they increase the prevalence of microplastics in soils. Numerous studies have demonstrated the link between using plastic mulch and the number of plastics detected in soil, as well as the favorable relationship between exposing the buildup of microplastics from this practice [69,70]. According to a report, endocytosis allowed polystyrene nano-beads (less than 100 nm) to penetrate tobacco cells [70]. Li et al. [16] found that polystyrene microplastics (0.2 m) can translocate through the soil to plant tissue cultures. The negative impacts were observed when the biodegradable and polyethylene microplastics were applied together to the crop plants [68,69]. Furthermore, they revealed that biodegradable microplastics can minimize fruit biomass and hinder the activity of earthworms in the soil. This study raised a new issue regarding the use of biodegradable plastics because some researchers encouraged

the use of biodegradable plastics in place of conventional plastics to reduce the issue of microplastic pollution.

**Table 3.** Microplastic estimation from the different farms input to the soil.

| Source | Country | Quantity | References |
|---|---|---|---|
| Mulching sheets | China | Topsoil 8885 particles per kg Deep subsoil 2899 particles per kg | [71] |
| Compost | Spain | 888 | [72] |
| Compost | China | 2400 | [73] |
| Fertilizers | Japan | 6–369 mg per kg | [74] |
| Polyhouse | China | 1000–3786 | [47] |
| Pig Manure | China | 43.8 | [75] |
| Sheep Manure | Spain | 997 | [76] |
| Mulching sheets | Spain | 2242 | [72] |
| Mulching sheets | Spain | 2116 | [76] |
| Mulching sheets | China | 80–308 | [69] |
| Mulching sheets | Republic of Korea | 215–3315 | [48] |
| Mulching sheets | China | 420–1290 | [38] |
| Mulching sheets | China | 900–2200 | [77] |
| Mulching sheets | China | 310–5698 | [78] |
| Sewage-Sludge | Canada | 541 | [79] |
| Sewage-Sludge | Netherlands | 5190 | [80] |
| Sewage-Sludge | China | 87.6–545.9 | [81] |
| Sewage-Sludge | Ireland | 4200–15,000 | [82] |
| Mulching sheets | China | 80.3–1075.6 particles per kg | [69] |
| Mulching sheets | China | 263–571 | [83] |
| Sewage-Sludge | Chile | 1100–3500 | [84] |
| Compost | Germany | 96 | [85] |
| Farms soil | USA | 100, 300, 500, and 700 nm | [54] |

In this section, we have discussed the various field inputs that lead to the transport of microplastics to soil and their possible impact on soil health. We have also presented the different channels of microplastic pollution in agricultural soils. The direct and indirect sources of microplastics have been summarized in Table 3.

## 6. Mitigation Strategies for Microplastic Pollution

Microplastic is an emerging issue for the terrestrial as well as for the aquatic ecosystem but the mitigation and adaption strategies play an important role in arising of new problems. Many strategies can be used to mitigate microplastic in general and particularly in the Farmer's field.

### 6.1. General Strategies against Microplastic Pollution

The general strategies can help to reduce the use and manufacturing of new plastic sources. These are the foremost strategies that can regulate of production and consumption of microplastics [86,87]. The new plastic should have an eco-design in nature that can follow the motto of the 4Rs i.e., reduce, reuse, recycle, and recover. After manufacturing, the reduction in the consumption of plastic products should be regulated by avoiding

unnecessary packaging. There should be increasing awareness regarding the use of microplastics in the environment, health, and food systems through formal or informal means. Education is a long-term goal that can help reduce the consumption and use of plastics. Increasing demand for plastic-free products, such as plates, spoons, and glasses, on the kiosks. The companies should be forced to redesign their products and make them from recycled plastics. There should be an improvement in plastic waste collection systems and proper landfills. The government should make a policy on the use of bio-based and biodegradable plastics and provide a tax rebate on recycled plastic. There should be an improvement in the recyclability of e-waste [87,88]. The long-term, mid-term, and short-term measures have been summarized in Figure 5.

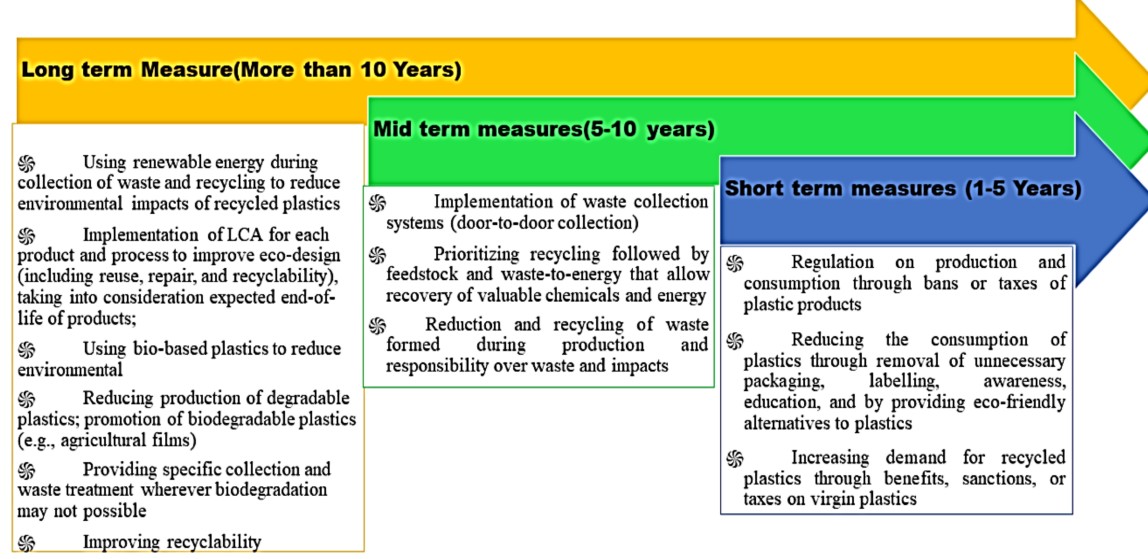

**Figure 5.** Classification of mitigation strategies based on time; three types of mitigation measures long-term, mid-term measures, and Short-term measures.

### 6.2. Specific Recommendation for Farmer Field

There are specific recommendations for the horticultural field to reduce and eliminate microplastic pollution. The proposed mitigation strategies have been described below

### 6.2.1. Use of the Biological Mulches

Plastic mulches are used in the horticultural field. These can be replaced with straw mulches such as paddy straw, wheat straw, and some long grasses, which can properly eliminate the use of plastic in the field. For crops such as strawberries, temperate crops and chilies, and capsicum, mulching is a term used in a broad sense.

### 6.2.2. Promotion of Natural and Organic Farming

The promotion of natural farming might reduce the use of coated fertilizers and other synthetic products that contain microplastics. The use of only own farm-based organic products in which the elimination.

### 6.2.3. Proper Check of the Entry in the Field of Sludge

In developed countries such as the USA, China, and the EU countries, sewage sludge is used as fertilizer and an irrigation source for growing crops. This is the major source of microplastic pollution, which can be reduced with the proper check of sludge entry and the recommended testing methods. This is the only way that it could potentially reduce the contamination of the field.

### 6.2.4. Use of Bioremediation by Biological Means

Soil and plant health play a crucial role in the proper growth and development of plants. But contamination of soil retards the proper function of the crop as well as the biota. Bioremediation of soil is the paramount way to certainly improve the ailment of the soil. In the bioremediation of soil, microbes feed on chemical pollutants and derivatives of microplastic by using metabolic mechanisms.

### 6.2.5. Use the Cemented and Biodegradable Pipes for Irrigation Channel

Nowadays, irrigation channels are plastic tubes, foldable pipes, and plastic pipes, and that can be the source of microplastic pollution because some particles may be removed during irrigation. So, the cemented and biodegradable pipes can be irrigation channels.

### 6.2.6. Reduce the Indiscriminate Use of Coated Fertilizer

The plastic-coated urea can be used for nutritional purposes, but the farming community used the indiscriminate form, which might have enhanced the chances of pollution. Proper use and the exact quantity of the fertilizers can be a solution to reduce microplastic pollution at the smaller end.

### 6.2.7. Use the Nanotechnology

Greenhouses and storage systems are made of plastic. The widespread usage of nanotechnology in horticulture is a problem since it replaces non-biodegradable materials in food production. Biopolymers such as chitosan are used in the production of green nanoparticles, whose manufacturing is simple, inexpensive, and biodegradable, which leads to the reduction in micro and nanoplastics in crop production systems [89]. The use of green nanoparticles is an emerging sector in horticulture such as fertilization of nutrients and insect pests and disease management [89–91]. However, no research is available on the production and use of green nanoparticles that lead to reducing the micro and nanoplastics in the farmer's field.

### 7. Conclusions

Based on formal deliberation, it is concluded that microplastic pollution on land continues to increase and threaten the health of humans and ecosystems. Nanoplastics are probably severely more hazardous to living beings than microplastics because of their greater abundance and reactivity. Food chain contamination and the identification of food safety issues may result from the buildup of nanoplastics in plants and animals. Microplastics could distress the crops at the cellular level and reduce metabolic activities, leading to genotoxicity and oxidative damage in horticultural crops. This may reduce the quality of the harvested commodity. As a result, the marketing value is reduced because it is unable to meet food safety standards. However, various indications have been shown that fruit and vegetables are contaminated by microplastics, which might have negative effects on living beings. Microplastics can be increased by the farm's input waste in the soil as well as microorganism's level. There is a proper requirement for regulations on microplastics in the short, medium, and long term, which helps reduce microplastic pollution in the horticulture sector.

**Author Contributions:** Conceptualization: U.S., S.S. (Sunny Sharma), V.S.R.; validation: U.S., S.S. (Sunny Sharma), N.R., and V.S.R.; resources and visualization: S.S. (Shilpa Sharma) and S.A.B.; writing—original draft preparation: S.S. (Shilpa Sharma); writing—review, and editing, H.Q., V.K. (Vijay Kumar), N.R., S.A.B., V.K. (Vineet Kumar), S.A.B.; supervision: V.K. (Vineet Kumar), U.S., S.A.B. and S.S. (Sunny Sharma). All authors have read and agreed to the published version of the manuscript.

**Funding:** This research received no specific grant from any funding agency in the public, commercial, or not-for-profit sectors.

**Institutional Review Board Statement:** Not applicable.

**Informed Consent Statement:** Not applicable.

**Data Availability Statement:** Not applicable.

**Acknowledgments:** The authors wish to thank Yashwant Singh Parmar University of Horticulture and Forestry, Solan and School of Agriculture, Lovely Professional University for providing necessary facility.

**Conflicts of Interest:** Authors declare that there is no conflict of interest in this study.

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
