# Peer review of "Assessment of Microplastics Pollution on Soil Health and Eco-toxicological Risk in Horticulture"

_soilsystems, doi:10.3390/soilsystems7010007_

Round 1
Reviewer 1 Report
L. 54-55: check the definition of microplastics < 5 mm
L115: biological photo-degradation caused by electromagnetic radiations pls. clarify this contradicting
statement
L. 120 123 The same statement is repeated
L138: specify the organisms
L. 146.147 Revise the sentence feed – food?
L164-166. This a very harsh statement. It should be very specific and not general.
Table 2. Pls specify the abbreviated techniques/technology; FTIR why do you add infrared after FTIR?
IRMS- ratio mass spectrometer Analysis: provide the correct description
279: human and animal health are not considered to be part of the environment
284: 34±0.36 microplastic derivatives per kg in topsoil pls. clarify the statement
289: pls revise this sentence
Labels in Figures 2-5 are blurred, they should be replaced with sharp text.

Author Response
Kindly see the attachment

Reviewer 2 Report
This review summarized the previous studies about microplastic pollution on soil health and ecotoxicological risk in horticulature. This topic seem really interesting, but the work have been less impressively done with lack of description and discussion. This munuscript could not be accepted in present form. Please check my comments and suggestion below.
1. The abstract not really describe overall of this work. Most of sentences explained why this review should be done and how this review is important. But the information obtained by the reviewing process did not enough explain as well as discussion and conclussions.
2. The pictures provided in this manuscript are too blurry to see clearly, please reset.
3. Line 79-80 'In the context of microplastic pollution, it is a rapidly emerging subject of nanotechnology research'. Are microplastics nanotechnology?
4. Section 4: The information about determination of microplastics needed to be summarized and discussed in details. The content of last paragraph was not belong to 'determination of microplastics'.
5. Why do some words start with capital letters in many sentences in the manuscript? please check.
6. There are relatively few references, and the cited papers are preferably from the last two years like 'Microplastic pollution in the soil environment: Characteristics, influencing factors, and risks' and 'Environmental source, fate, and toxicity of microplastics'.
Author Response
Kindly see the attachment

Reviewer 3 Report
The authors present a review on microplastics pollution in soil and horticulture. Overall, this review is timely and useful for our understanding of microplastics in agriculture system as well as in horticultural crops. However, there are many issues need to be addressed as follows.
Line 43-45. “In 2019, plastic production worldwide was 368 Mt where China contributes 114 Mt followed by Europe (59 Mt),……… [2].” Reference [2] is Sci. Adv 2017, 3(7), e1700782. How can the reference published in 2017 draw the conclusion of the data in 2019?
Line 47-49. “Researchers have investigated that how microplastic particles go to the food chain and are consumed by humans, potentially endangering their health”. Need citations.
Line 52-53, “The existence of microplastics from various sources in the terrestrial ecosystem has recently become a hotspot of current research [3].” Reference [3] is about personal care products, here need more citations about terrestrial ecosystem.
Line 53-54, “ MPs; > 5mm” should be “MPs; <5mm”
Line 61-64, reference [8] is “Microplastic characteristic in the soil across the Tibetan Plateau” unrelated to the meaning of the sentence in line 61-64.
Line 68-69, add a more recent reference to [9-10]. Environmental microplastics: Classification, sources, fates, and effects on plants. Chemosphere, 2023, 313, 137559
Line 81-82, what does “fruit plant cells” mean?
Line 82, 85, 86. “Microplastics” should be “microplastics”. Check “Microplastics” throughout the manuscript.
Line 117-118, “In the ocean, the secondary microplastic is detected of a size of 1.6 micrometres in diameter [18].” However, reference [18] is “Mediterranean diet and prevention of chronic diseases.” Is it [19]?
Line 119-125, this paragraph is confusing logically and needs to be rewritten.
Line 131-132, “Spectroscopy is a novel technique for the identification and quantification of nanoplastics [24-25].” Add a more recent reference to [24-25] in novel technique for the identification and quantification of nanoplastics. Indoor microplastics and bacteria in the atmospheric fallout in urban homes. Science of The Total Environment, 2022, 852, 158233.
Line 135-137. ”The study showed that the nanoplastics have surpassed the epithelial membrane of fish and microparticles get deposited into various organs namely, the gall bladder, pancreas, and the brain [26].” However, reference [26] is “Desalination techniques—A review of the opportunities for desalination in agriculture.”
Line 138-140, please check reference [27-28] whether they are related to “In other organisms……”.
Line 162, “Peoples” should be “peoples”.
Line 166-169, need a citation.
Line 223-224,”Horticultural crops are herbaceous and perennial due to the more favourable to contamination by microplastics.” This sentence is problematic.
Line 225, there are two “therefore”.
Table 1. [45] Dong et al. 2021.
Line 268, “composition” should be “composition.”
Line 261-273, a few studies of positive effect of microplastics should not be ignored, and should be fairly commented.
Line 285-287, “In this section, we have discussed the various field inputs that lead to the transport of microplastics to soil and their possible impact on soil health. We have also presented the different channels of microplastic pollution in agricultural soils. The direct and indirect sources of microplastics have been summarized in table 3.” Move them to the end of this paragraph or delete them.
Line 345, change the (Bandmann et al. 2012) to number citation.
Line 363-364, “The general strategies which can help to reduce the use and manufacturing of new plastic sources.” Delete the “which”.
Line 429-430, “Hence reducing the export potential because it is not able to qualify for the food safety standards.” What does “export potential” mean? Need to be reworded.
Author Response
Kindly see the attachment

Round 2
Reviewer 2 Report
The reviewers remarks have been taken into consideration, and the revised text may be accepted for publication.
Reviewer 3 Report
All my questions and concerns have been addressed.